# Intracranial Haemorrhage in Haemophilia Patients Is Still an Open Issue: The Final Results of the Italian EMO.REC Registry

**DOI:** 10.3390/jcm11071969

**Published:** 2022-04-01

**Authors:** Ezio Zanon, Samantha Pasca, Francesco Demartis, Annarita Tagliaferri, Cristina Santoro, Isabella Cantori, Angelo Claudio Molinari, Chiara Biasoli, Antonio Coppola, Matteo Luciani, Gianluca Sottilotta, Irene Ricca, Berardino Pollio, Alessandra Borchiellini, Alberto Tosetto, Flora Peyvandi, Anna Chiara Frigo, Paolo Simioni

**Affiliations:** 1Haemophilia Center-General Medicine, Padua University Hospital, 35128 Padua, Italy; paolo.simioni@unipd.it; 2Department of Biomedical Sciences, Padua University Hospital, 35128 Padua, Italy; sampasca27@gmail.com; 3Centre for Bleeding Disorders, Careggi University Hospital of Florence, 50134 Florence, Italy; fdemartis@amm.unica.it; 4Regional Reference Centre for Inherited Bleeding Disorders, University Hospital of Parma, 43126 Parma, Italy; annaritatag@gmail.com (A.T.); ancoppola@ao.pr.it (A.C.); 5Hematology Division, Umberto I University Hospital of Rome, 00185 Rome, Italy; santoro@bce.uniroma1.it; 6Haemophilia Center, Department of Transfusion Medicine, Hospital of Macerata, 62100 Macerata, Italy; isacantori@libero.it; 7Regional Reference Centre for Hemorrhagic Diseases, Thrombosis and Hemostasis Unit, Gaslini Children Hospital of Genoa, 16147 Genova, Italy; aclaudiomolinari@gaslini.org; 8Haemophilia Center, Transfusion Medicine, Department of Clinical Pathology, Hospital of Cesena, 47521 Cesena, Italy; chiara.biasoli@auslromagna.it; 9Haemostasis and Thrombosis Center, Onco-Hematology Department, Bambin Gesù Children Hospital of Rome, 00165 Roma, Italy; matteo.luciani@opbg.net; 10Haemophilia Center, Department of Onco-Hematology and Radioterapy, Hospital of Reggio Calabria, 89124 Reggio Calabria, Italy; gianluca.sottilotta@virgilio.it; 11Transfusion Medicine, Department of Diagnostic, Regina Margherita Children Hospital of Turin, 10126 Turin, Italy; iricca@cittadellasalute.to.it (I.R.); pollio.berardino@gmail.com (B.P.); 12Hemostasis and Thrombosis Unit, Molinette Hospital of Turin, 10126 Turin, Italy; aborchiellini@cittadellasalute.to.it; 13Hemorrhagic and Thrombotic Diseases Unit, S. Bortolo Hospital of Vicenza, 36100 Vicenza, Italy; alberto.tosetto@aulss8.veneto.it; 14Hemophilia and Thrombosis Center, University Hospital of Milan, 20132 Milan, Italy; flora.peyvandi@unimi.it; 15Department of Cardiac, Thoracic, Vascular Sciences and Public Health Padua University Hospital, 35122 Padova, Italy; annachiara.frigo@unipd.it

**Keywords:** intracranial haemorrhage, haemophilia A and B, risk factors, incidence, mortality

## Abstract

Background: Intracranial hemorrhage (ICH) is a highly serious event in patients with haemophilia (PWH) which leads to disability and in some cases to death. ICH occurs among all ages but is particularly frequent in newborns. Aim: The primary aim was to assess the incidence and mortality due to ICH in an Italian population of PWH. Secondary aims were to evaluate the risk factors for ICH, the role of prophylaxis, and the clinical management of patients presenting ICH. Methods: A retrospective-prospective registry was established in the network of the Italian Association of Haemophilia Centers to collect all ICHs in PWH from 2009 to 2019 reporting clinical features, treatments, and outcomes. Results: Forty-six ICHs were collected from 13 Centers. The ICHs occurred in 15 children (10 < 2 years), and in 31 adults, 45.2% of them with mild hemophilia. Overall, 60.9% patients had severe haemophilia (15/15 children). Overall ICH incidence (×1000 person/year) was 0.360 (0.270–0.480 95% CI), higher in children <2 years, 1.995 (1.110–3.442 95% CI). Only 7/46 patients, all with severe haemophilia, had received a prophylactic regimen before the ICH, none with mild. Inhibitors were present in 10.9% of patients. In adult PWHs 17/31 suffered from hypertension; 85.7% of the mild subjects and 29.4% of the moderate/severe ones (*p* < 0.05). ICH was spontaneous in the 69.6% with lower rate in children (46.7%). Surgery was required in 21/46 patients for cerebral hematoma evacuation. Treatment with coagulation factor concentrates for at least three weeks was needed in 76.7% of cases. ICH was fatal in 30.4% of the cases. Of the survivors, 50.0% became permanently disabled. Only one-third of adult patients received long term prophylaxis after the acute treatment. Conclusion: The results from our Registry confirm the still high incidence of ICH in infants <2 years and in adults, particularly in mild PWHs presenting hypertension and its unfavorable outcomes. The majority of PWHs were treated on-demand before ICH occurred, suggesting the important role of prophylaxis in preventing such life-threatening bleeding.

## 1. Background

Intracranial hemorrhage (ICH) is the most serious bleeding event in patients with haemophilia (PWH), leading to disability and in some cases to death. ICH occurs among all ages [1] but is more frequent in newborns and in adults as reported in different studies and reviews [2,3,4,5,6].

High incidence of cerebral bleeding in babies is often associated with mode of delivery, although this statement is still a subject of discussion among clinicians [7,8]. Usually, primary prophylaxis is established in children within the first years of age and the ICH often occurs in young patients who have not yet undergone this type of treatment. The role of prophylaxis is also shown in adult patients with different degrees of haemophilia in which severe haemorrhagic events occur prevalently in subjects only treated on-demand with coagulation factor concentrates or with poor adhesion to a prophylaxis regime [9,10]. The incidence of mortality has been estimated at 0.8 per 1000 persons per year [1], 3.5 times greater than the general population [11].

The severity of haemophilia is the most relevant risk factor for ICH in children and adolescents [9,11], while in adults intracranial bleeding can also occur in mild or moderate patients, especially in those who present concomitant viral infections (e.g., HIV or HCV) or uncontrolled hypertension [12,13]. In 2012 we published [3] a retrospective survey (1987–2008) showing that in an Italian population of 88 patients with a total of 112 ICH events, the related risk factors for ICH were: age <2 years and over >50 years, the severity of haemophilia, the presence of inhibitors and hepatitis C virus (HCV) infection, on-demand treatment, alongside hypertension in cases of mild PWH. The mortality rate related to intracranial bleeding was high, especially in the patients with inhibitors (50%), while 22.7% of patients who survived had disabling sequelae.

From this background a new study was established (EMO.REC registry) which collected data about ICH in a population of PWH from 13 Italian Haemophilia Centers from 2009 to 2019 to verify whether over the last 10 years the newly introduced therapies and the diffusion of prophylaxis have changed the epidemiology and characteristics of such serious complications.

## 2. Aim

The EMO.REC retrospective-prospective registry was primarily intended to assess the incidence and mortality due to ICH in a population of Italian patients with haemophilia A and B.

The secondary objectives included the evaluation of the role of risk factors involved in ICH occurrence, the role of prophylaxis to prevent acute bleeding and recurrences, and the clinical management of patients presenting ICH.

## 3. Patients and Methods

### 3.1. Patients

The EMO.R.EC study was proposed and approved in the frame of network of the Italian Association of Haemophilia Centers (AICE). Thirteen Centers contributed to the Registry, with enrollment of retrospective and prospective PWHs with ICH starting in October 2012.

Patients with combined or acquired coagulation disorders were excluded from this registry.

Retrospective group: all patients with haemophilia A or B, of any age, presenting ICH over three years (from January 2010 to September 2012).

Prospective group: all consecutive patients with haemophilia A or B, of any age, presenting ICH from October 2012 to December 2019.

The study protocol was approved by each institution’s Ethical Committee and was conducted in accordance with the principles of the Declaration of Helsinki and with local laws and regulations. All patients provided written informed consent. In the case of patients who died before the start of the data collection, the informed consent was not necessary according to the current legislation at the time.

### 3.2. Methods

All patients were assessed for (1) demographic and baseline characteristics such as age at diagnosis, laboratory and clinical conditions, co-morbidities; (2) descriptive characteristics of bleeding episodes (symptoms, site, possible cause); (3) acute treatment of ICH (time, dosage, ancillary therapy); (4) outcomes (mortality, disability); (5) long-term prophylaxis and recurrences; (6) any adverse event related or not to treatments.

According to International Guidelines [14], haemophilia A or B was classified as severe if the plasma factor VIII/IX level was <1%, moderate if 1 to <5% and mild if 5–30%.

Inhibitor patients were defined as those having an inhibitor titer > 0.6 BU on at least two separate determinations, the inhibitor titer being measured in each participating center with the Bethesda assay modified Nijmegen.

An intracranial haemorrhage was diagnosed or confirmed with radiological investigations. (e.g., CT-scan or magnetic resonance imaging or ultrasound scanning in cases of newborns).

Recurrences were defined as a haemorrhagic cerebral event occurring in the same site more than one month from the first bleeding.

Prophylaxis was defined as a regular infusion of a clotting factor concentrate for at least 46 weeks/each year.

### 3.3. Statistical Analysis

Statistical analyses were performed with SAS 9.4 (SAS Institute Inc., Cary, NC, USA) for Windows. All the variables collected were summarized in tables by appropriate descriptive statistics: mean, standard deviation (SD), median, count and percent.

AEs were coded using the MedDRA (Medical Dictionary for Regulatory Activities).

ICH free survival probability in the different haemophilia severity groups (mild/moderate/severe) was represented graphically with Kaplan-Meier curves and compared with the log-rank test adjusting the *p*-values with the Bonferroni’s method. ICH incidence with 95% confidence interval (CI) was estimated considering a Poisson model.

Results are considered statistically significant when *p*-value is <0.05

## 4. Results

Statistical analysis included all data of the 46 ICH enrolled patients (29 patients in the prospective group and 17 in the retrospective one), from a population of 3077 haemophiliacs treated at the 13 Haemophilia Centers contributing to the EMO.REC.

15/46 patients were children or adolescents (age ≤ 16 years), all presenting severe haemophilia. Among the remaining 31 adults, 41.9% presented severe haemophilia A or B, 12.9% were moderate and 45.2% mild. Median age at ICH diagnosis was 1.0 years (range birth-14) in children, 53.0 years (range 17–92) in adults, being higher in those with mild haemophilia, 62.5 years (range 43–92) than in the severe/moderate ones, 45.0 (range 17–76) *p* < 0.22. Mild adult patients were 34.5% in the prospective group, while 23.5% in the retrospective one; median age was lower in the first group. Family history for haemophilia A or B was found in 41.3% of patients, and 33.4% in children. Genetic analyses were available for 32/46 patients, among 27 patients with haemophilia A 59.9% had an inversion of intron 22. Inhibitors were positive at the ICH onset in 5/46 patients, only one with haemophilia B; three were children; one had mild haemophilia A. Previous viral infection was found in 77.4% of the adults, 95.8% of them were HCV+, while 60.9% two or more cases had combined infections. 3 Among adult patient 32.2% were smokers, equally divided between mild and severe-moderate patients, while 58% used alcohol habitually or occasionally. Complete data of enrolled patients are reported in Table 1.

ICH was a spontaneous event in overall 69.6% of cases with higher proportions in adult mild patients (92.9%), than in the severe-moderate group (70.6%). Conversely in the children, in which 53.3% had a traumatic ICH in a quarter of them due to delivery. More than half (56.5%) presented one or more concomitant disease, of which the most frequent was hypertension. The detail of the comorbidities in adults is shown in Figure 1.

At the ICH onset only 8/46 (17.4%) patients were on prophylaxis. All had severe haemophilia A (two children, six adults) and were treated with standard half-life FVIII concentrates. Seven children <1 years of age had not yet started any treatment. One haemophilia A child had the haemorrhagic event during the Immune Tolerance Induction (ITI) regimen. Symptoms, sites, and outcomes of ICH are reported in Table 1.

### Prophylaxis

Neurosurgical interventions for evacuation of the cerebral haematoma were carried out in 45.7% of cases while 76.7% of patients received treatment with factor FVIII, FIX or rFVIIa for at least three weeks from the ICH onset. Eight patients died immediately or within five days from the haemorrhagic event. For three patients, data regarding the treatment in the days following admission were missing and only their ongoing therapy was reported.

ICH incidence calculated for the different ages is reported in Table 2.

Overall ICH incidence (×1000 person/year) was 0.360 (0.270–0.480 95% CI), higher in children <2 years, 1.995 (1.110–3.442 95% CI).

Figure 2 reports the Kaplan Mayer curves which describe survival probabilities without ICH in patients with different severities. Patients with severe haemophilia A and B show higher risk of ICH than those with moderate to mild haemophilia.

High mortality was found in the adults (41.9%), especially in those with severe-moderate disease (52.9%), where it was almost twice than of mild patients (28.5%). Subjects who died before starting or during treatment were 31.8%. Only one child 18 months old with severe haemophilia B died due to ICH complications.

50% of the survivors had a permanent disability without any difference among the different forms of haemophilia.

Survival curves for different degrees of haemophilia are reported in Figure 2.

Patients treated with FVIII/FIX concentrates or rFVIIa for at least three weeks were 76.7%.

Long-term prophylaxis was maintained among survivors in only approximately one third of adult patients (38.0%), higher in the severe-moderate (6/8) than in the mild disease (2/13) cases. Half of the children continued a prophylactic treatment, one child continued the ITI already started before the bleeding episode, while one started it after the acute event was resolved. Another child was switched to emicizumab after acute treatment with rFVIIa.

## 5. Discussion

Despite the remarkable progress achieved in the treatment of haemophilia, intracranial hemorrhage continues to be the most severe haemorrhagic complication in haemophiliac patients. Furthermore, the mortality remains high in these patients [1,15]; therefore, it is a leading cause of death in haemophiliacs [16].

Intracranial haemorrhage may occur at different ages, with a peak in newborns, reducing in infants and adolescents, and resuming in adults over 40 years [3]. The risk of an ICH development in haemophiliacs, both as a whole and in pediatric patients, remains higher than in the general population [1].

Also, the mortality rate, estimated at 0.8 out of 1000 persons per year, is higher in the general population [11].

Intraparenchymal and subarachnoid spontaneous forms prevail while the subdural haematoma is usually post traumatic and more frequent in pediatric patients [3,12,17,18]. Also, in our study intraparenchymal haemorrhage is the site most affected (51.6% of the adults and in 33.4% of the pediatrics). Subdural haematoma is more common in children, in 40% of the pediatric traumas it was documented.

The ICH incidences reported in all published studies are 2.3 events per 1000 patients per year considering all ages [1] similarly to what was reported in our previous study of a cohort from 1989 to 2005 in which the result was 2.5 events per 1000 patients per year [3].

In this study we report an annual rate of ICHs for the whole cohort over the entire follow-up period of 0.36 events per 1000 patients (95% CI: 0.27–0.48). When patients with severe haemophilia are considered, in the most recent cohort the incidence of ICH was 0.56 events per 1000 patients against 3.40 reported in the previous cohort [3]. This is a much lower incidence rate in respect to previous studies [2,4,5,12], but the observation period is more recent and covers a period between 2009 and 2019.

The wider adoption of the prophylaxis in patients of all ages affected by severe haemophilia in the last decade could probably explain these different results, as previously reported [9,10]. Further confirmation of this may be the observation of a high incidence of ICH in patients in their first two years of life, 1955 per 1000 patients per year, i.e., an early age at which prophylaxis with a substitution factor has not been completely implemented yet.

Although having severe haemophilia is confirmed as the greatest risk for a patient to suffer from a cerebral haemorrhage [3,6,10,19], which is at variance with previous studies, in this registry we found an elevated percentage mild haemophiliacs among ICH patients. 

Mild haemophilia was present in 14 out of 31 adults. The mean age of patients with mild haemophilia was 17.5 years higher than severe patients (mean 62.5 years in the mild cohort versus 45.0 years in the combined moderate and severe cohort). In 91% of the cases ICHs were spontaneous, higher than in the other ones (75%).

Hypertension was present in 11 out of 14 mild patients (78.6%), while only 33.3% of non-mild subjects had this co-morbidity (*p* < 0.05). Among patients with diagnosed hypertension, 88.9% were in treatment with anti-hypertensive drugs, but without an effective control of the blood pressure values. This should not surprise us since it has been seen that hypertension is more frequent in persons with haemophilia than control groups, particularly in 50–59-year-olds [13,20]. Checking for hypertension during a routine examination is seriously recommended by the recently published World Federation of Hemophilia guidelines [21].

All mild patients were treated with coagulation factor concentrates only on-demand at the time of ICH; the same was also reported in 75% of patients with moderate to severe types, but while the children with severe haemophilia who experienced intracranial hemorrhage were not on prophylaxis because it had not yet been started at the time of the event due to their young age, the adults with moderate and severe haemophilia had chosen not to do a prophylaxis regime despite having been invited to do so by their hematologists. The results of our study would therefore seem to confirm the protective role of prophylaxis in preventing ICH.

The mortality rate for ICH even if it is double in severe patients also seems to be a relevant cause of death in mild patients as previously reported, 12% in a cohort of 2079 patients [10] Permanent disability is present in 50% of patients in both severe and mild haemophiliac patients.

The data on prophylaxis after such severe haemorrhagic events are not reassuring.

Prophylaxis after the acute treatment was continued life-long only in 25% and 9% of the patients with severe and mild haemophilia respectively, none of them subsequently developed inhibitors.

From our results therefore it is evident that comorbidities, in particular hypertension and the treatment modality play a key role in the epidemiology of ICH in haemophilia.

Although demonstrating a reduction in incidences of ICH over the last decade, this study also shows that ICH may occur in all patients with all severities of haemophilia treated on-demand and in adults with comorbidities.

It is urgent and necessary to launch new therapeutic strategies in order to reduce the impact of such life-threatening and disabling events in the haemophiliac population as much as possible.

Based on these data our advice would be:Start prophylaxis (and carefully monitor adequate adherence) in all patients affected by severe hemophilia but also in those with mild forms in whom risk factors (hypertension in particular) are effectively controlled.Check the risk factors and treat the comorbidities of the patient during follow-up examination.In newborns with severe forms, start a prophylaxis treatment as soon as possible.

The availability of FVIII and FIX extended half-life concentrates on the market which reduces the number of infusions and of replacement therapies which are easy to administer subcutaneously, currently provide advantages in convincing the patient to comply with the treatment. Therefore, these novel therapeutic approaches can facilitate starting a prophylaxis regimen in order to prevent cerebral haemorrhages and possible recurrences.

The current possibility of starting a non-replacement therapy from birth with emicizumab for patients with haemophilia A, as advised by the guidelines of the WFH [21], and in the future perhaps also with other drugs (studies are ongoing) in patients with haemophilia B, could change the epidemiology of such manifestations in newborns in whom, however, trauma and delivery are still relevant challenges.

Ongoing studies are addressing the efficacy and safety of a very early start of prophylaxis with emicizumab in children <1 year. Their results will be likely to provide some answers about the possibility of improving the prevention of such serious complications.

## Figures and Tables

**Figure 1 jcm-11-01969-f001:**
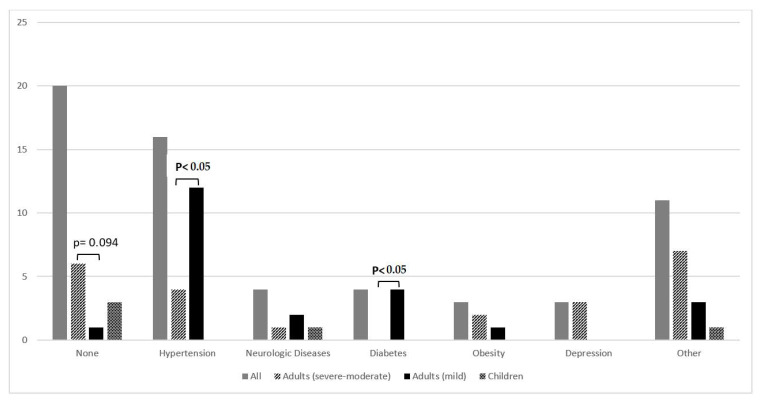
Concomitant diseases reported at ICH onset. 11 of 26 patients presenting other diseases had two or more comorbidities.

**Figure 2 jcm-11-01969-f002:**
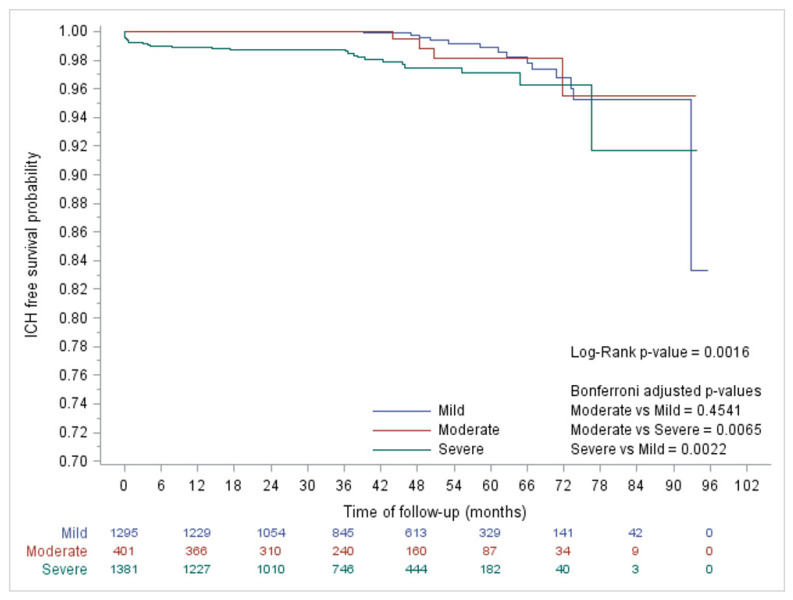
Intracranial haemorrhage (ICH) free survival probability in the different haemophilia degrees.

**Table 1 jcm-11-01969-t001:** Baseline characteristics of patients at intracranial haemorrhage diagnosis. Symptoms, sites and outcomes of intracranial haemorrhage (ICH). * Patients can have one or more symptoms or ICH sites. NA: not applicable; data referred only to adult patients (*n* = 31).

	All *n* (%)	Adults *n* (%)	Children *n* (%)
**Number of total patients**	46 (100.0)	31 (100.0)	15 (100.0)
**Type of haemophilia:**			
A	39 (84.8)	26 (83.9)	13 (86.7)
B	7 (15.2)	5 (16.1)	2 (13.3)
**Degree of haemophilia:**			
Mild	14 (30.4)	14 (45.2)	
Moderate	4 (8.7)	4 (12.9)	
Severe	28 (60.9)	13 (41.9)	15 (100.0)
**Family history:**			
Family	19 (41.3)	14 (45.2)	5 (33.4)
Sporadic	27 (58.7)	17 (54.8)	10 (66.6)
**Inhibitor:**			
Yes	5 (10.9)	2 (6.5)	3 (20.0)
No	41 (89.1)	29 (95.5)	12 (80.0)
**Viral Infections:**			
None	22 (47.8)	7 (22.6)	15 (100.0)
HCV	23 (50.0)	23 (74.2)	
HIV	3 (6.5)	3 (9.7)	
HBV	2 (4.3)	2 (6.5)	
Other	1 (2.2)	1 (3.2)	
**Trauma:**			
No	32 (69.6)	25 (80.6)	7 (46.7)
Yes	12 (26.1)	6 (19.4)	6 (40.0)
Peri-Partum	2 (4.3)		2 (13.3)
**Smoke:**			
Yes	10 (32.3)	10 (32.3)	NA
No	17 (58.8)	17 (58.8)	NA
Not reported	4 (12.9)	4 (12.9)	NA
**Alcohol intake:**			
No	11 (35.5)	11 (35.5)	NA
Usually	13 (41.9)	13 (41.9)	NA
Rarely	3 (9.7)	3 (9.7)	NA
Not reported	4 (12.9)	4 (12.9	NA
*** Symptoms at diagnosis:**			
None	13 (28.2)	11 (35.5)	2 (13.3)
Headache	12 (26.0)	9 (29.0)	3 (20.0)
Epilepsy	4 (8.7)	1 (3.2)	3 (20.0)
Motor dysfunction	4 (8.7)	3 (9.7)	1 (6.7)
Syncope	1 (2.2)	1 (3.2)	
Asthenia	5 (10.9)	4 (12.9)	1 (6.7)
Drowsiness	7 (15.2)	4 (12.9)	3 (20.0)
Coma	7 (15.2)	5 (16.1)	5 (33.4)
Vomiting	2 (4.3)	1 (3.2)	1 (6.7)
Other	3 (6.5)	3 (9.7)	
*** Site of ICH:**			
Intraparenchymal	21 (45.7)	16 (51.6)	5 (33.4)
Subdural	8 (17.4)	4 (12.9)	4 (26.7)
Subarachnoid	9 (19.6)	6 (19.4)	3 (20.0)
Intraventricular	2 (4.3)		2 (13.3)
Not specified	11 (23.9)	8 (25.8)	3 (20.0)
**Sequelae:**			
All	13 (28.2)	8 (25.8)	5 (33.4)
Neurological	5 (10.9)	2 (6.5)	3 (20.0)
**Recurrences**	2 (4.3)	1 (3.2)	1 (6.7)
**Deaths**	14 (30.4)	13 (41.9)	1 (6.7)

**Table 2 jcm-11-01969-t002:** Intracranial haemorrhage (ICH) incidence in the different age groups.

	ICH *n* (%)	Years at Risk	Rate (×1000/Person Years)	95% CI
**Whole population**	46 (100.0)	127,923.5	0.360	0.269–0.480
**0–2 years**	12 (26.1)	6139.0	1.955	1.110–3.442
**3–16 years**	4 (8.7)	46,754.0	0.086	0.032–0.228
**17–40 years**	7 (15.2)	95,018.5	0.074	0.035–0.155
**41–59 years**	12 (26.1)	85,585.0	0.140	0.080–0.247
**≥60 years**	11 (23.9)	42,348.9	0.260	0.144–0.469

## Data Availability

The data that support the findings of this study are available on request from all authors. The data are not publicly available due to restrictions e.g., since it contains information that could compromise the privacy of research participants.

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
