# Peer review of "Intracranial Haemorrhage in Haemophilia Patients Is Still an Open Issue: The Final Results of the Italian EMO.REC Registry"

_jcm, 2022, doi:10.3390/jcm11071969_

Round 1

Reviewer 1 Report

Mild haemophilia was present in 11 out of 23 adults. The mean age of these patients was 45 years lower than mild patients (62.5 years). In 91% of the cases ICHs were spontaneous, higher than in the other ones (75%).

 Dear Author , 

A meaningful retrospective cohort study trying to answer a clinical question which may be life saving for many more patients 

Please do address this error appropriately. It sounds conflicting 

"Mild haemophilia was present in 11 out of 23 adults. The mean age of these patients was 45 years lower than mild patients (62.5 years). In 91% of the cases ICHs were spontaneous, higher than in the other ones (75%)."

Author Response

"Mild haemophilia was present in 11 out of 23 adults. The mean age of these patients was 45 years lower than mild patients (62.5 years). In 91% of the cases ICHs were spontaneous, higher than in the other ones (75%)."

  • The sentence has been changed in:

Mild haemophilia was present in 11 out of 23 adults. The mean age of these patients was 62.5 years higher than severe patients (45 years). In 91% of the cases ICHs were spontaneous, higher than in the other ones (75%).

Reviewer 2 Report

Thank you for asking me to review this paper looking at patients with haemophilia and intracranial haemorrhage rates compared to a retrospective cohort.  This is a valuable addition to the literature.  I have the following comments to make:

  1. It would be helpful to have more information on why some of the prospective cohort with severe haemophilia (and moderate haemophilia) were not on prophylaxis - had they been offered it and declined it, or were there local practices in place that meant it was not offered to them?
  2. Does the y axis of figure 1 represent percentage or absolute number?
  3. Is there any data on whether the patients with mild Haemophilia who went on prophylaxis after an ICH developed inhibitors?  I think this would be a clinical concern in this group and any data on this from this cohort or referenced from other studies would be very helpful.
  4. Of the patients in the prospective cohort who were on prophylaxis - what kind of prophylaxis was given (emicizumab / standard half life FVIII / EHLs)?

Author Response

Reviewer 2

It would be helpful to have more information on why some of the prospective cohort with severe haemophilia (and moderate haemophilia) were not on prophylaxis - had they been offered it and declined it, or were there local practices in place that meant it was not offered to them?

  • An explanation as to why these patients were not on prophylaxis was added to the text

Does the y axis of figure 1 represent percentage or absolute number?

  • Absolute number. An explanation has been included in the legend.

Is there any data on whether the patients with mild Haemophilia who went on prophylaxis after an ICH developed inhibitors?  I think this would be a clinical concern in this group and any data on this from this cohort or referenced from other studies would be very helpful.

  • None of the patients treated prophylaxis after ICH subsequently developed inhibitors. This sentence has been added to the text

Of the patients in the prospective cohort who were on prophylaxis - what kind of prophylaxis was given (emicizumab / standard half life FVIII / EHLs)?

  • All patients previous in prophylaxis were treated with standard half-life FVIII concentrates. This information has been added to text